# Supplier-origin mouse microbiomes significantly influence locomotor and anxiety-related behavior, body morphology, and metabolism

Aaron C. Ericsson [1,2✉], Marcia L. Hart[3], Jessica Kwan[4], Louise Lanoue[5], Lynette R. Bower[5], Renee Araiza [5,6], K. C. Kent Lloyd [5,6,7,8] & Craig L. Franklin [1,2✉]

The mouse is the most commonly used model species in biomedical research. Just as human physical and mental health are influenced by the commensal gut bacteria, mouse models of disease are influenced by the fecal microbiome (FM). The source of mice represents one of the strongest influences on the FM and can influence the phenotype of disease models. The FM influences behavior in mice leading to the hypothesis that mice of the same genetic background from different vendors, will have different behavioral phenotypes. To test this hypothesis, colonies of CD-1 mice, rederived via embryo transfer into surrogate dams from four different suppliers, were subjected to phenotyping assays assessing behavior and physiological parameters. Significant differences in behavior, growth rate, metabolism, and hematological parameters were observed. Collectively, these findings show the profound influence of supplier-origin FMs on host behavior and physiology in healthy, genetically similar, wild-type mice maintained in identical environments.

[1] Department of Veterinary Pathobiology, College of Veterinary Medicine, University of Missouri Metagenomics Center (MUMC), University of Missouri (MU), Columbia, MO, USA. [2] MU Mutant Mouse Resource and Research Center (MU MMRRC), Columbia, MO, USA. [3] IDEXX BioAnalytics, Columbia, MO, USA. [4] School of Veterinary Medicine, University of California (UC), Davis, CA, USA. [5] Mouse Metabolic Phenotyping Center (MMPC) at UC Davis, Davis, CA, USA. [6] Mutant Mouse Resource and Research Center at UC Davis, Davis, CA, USA. [7] UC Davis Mouse Biology Program (MBP), Davis, CA, USA. [8] Department of Surgery, School of Medicine, UC Davis, Sacramento, CA, USA. ✉email: ericssona@missouri.edu; franklinc@missouri.edu

Laboratory mice are the most commonly used model species in biomedical research, resulting in an ever-expanding corpus of knowledge encompassing almost every conceivable aspect of murine biology and physiology. As different inbred strains and outbred stocks were initially developed and distributed among different investigators in the early and mid-20th century, it became apparent that each line of mice had characteristic behavioral phenotypes[1–3], as well as physiological parameters[4], presumably due to the specific alleles selected and fixed in that specific strain or stock. As different inbred strains were established and characterized, they were also shared between labs and maintained as separate inbred breeding lines at different institutions. This practice resulted in the development of substrains if newly created inbred strains were separated between generations 20 and 40 or maintained for more than 20 subsequent generations of filial mating. Within C57BL/6 mice for example, the genetic differences between substrains from different suppliers have been documented[5,6], and there exist numerous reports of different substrain-dependent behavioral phenotypes[7–14], response to pharmacological stimuli[15–19], and disease susceptibility in models of metabolic and cardiac function and disease[20–28].

While host genetics explain a portion of these phenotypic characteristics, several lines of evidence indicate that the host-associated microbiota can also influence the host phenotype, in terms of behavior, metabolism, and physiology. Notably, germ-free (GF) mice display increased motor activity and reduced anxiety-related behavior relative to mice colonized with a standard specific pathogen-free (SPF) microbiota[29–31], and resistance or susceptibility to anxiety-related behavior can be transferred between SW and BALB/c mice via reciprocal fecal microbiome transfer (FMT)[32,33]. Similarly, GF mice require a significantly greater caloric intake than SPF mice to meet metabolic demands, have poorly developed immune systems, and other physiological abnormalities compared to their SPF counterparts. The term SPF however, is a definition of exclusion, and the relative influence of the different supplier-origin microbiomes on all of these parameters is unclear.

To investigate the influence of supplier-dependent microbiota in mouse models, four colonies of outbred CD-1 mice, each harboring a distinct fecal microbiota (FM) derived from mice purchased from the four main suppliers of laboratory mice in the U.S. were generated at the MU MMRRC. These colonies were established via embryo transfer (ET) rederivation of Crl:CD1(ICR) (referred to hereafter as CD-1) embryos in pseudopregnant C57BL/6J, C57BL/6NTac, CD-1, and C57BL/6NHsd surrogate dams purchased from Jackson, Taconic, Charles River, and Envigo (formerly Harlan), respectively. The resulting offspring, colonized naturally with the four different supplier-origin FMs, have been maintained as separate colonies for over 30 generations, retaining and vertically transmitting each FM from generation to generation, and remaining disparate when mice are shipped to and maintained at collaborating institutions[34]. To maintain these colonies, inbreeding is avoided, host genetics are 'refreshed' annually via the introduction of newly purchased stock using ET, and the FM is monitored quarterly to assess consistency within, and differences between, the various FM profiles.

Over multiple generations, FM-dependent behavioral differences were noted by animal care staff and principal investigators. To assess the effects of these naturally occurring FMs on behavior and basic physiology in genetically matched individuals, comprehensive phenotyping was performed using age-matched male and female CD-1 mice ($n = 8$/sex/FM) colonized with four FMs designated FM1 (Jackson), FM2 (Taconic), FM3 (Charles River), and FM4 (Envigo). Here we report the significant effect of these supplier-origin FMs on host behavior, body morphology, cardiac force, metabolism, and basic hematological parameters,

underscoring the relevance of these supplier-origin FMs in virtually any mouse model.

## Results

**Distinct supplier-origin FM profiles are maintained.** Fecal samples collected from each colony revealed conserved FM profiles within each of the four colonies, with distinct differences in both richness and β-diversity between colonies. Using samples collected at necropsy, two-factor ANOVA detected significant differences in richness between FMs ($p < 0.001$, $F = 10.1$), with no significant effect of sex or interactions, in the mice used in phenotyping tests (Fig. 1a). Pairwise comparisons indicated that FM1 was significantly less rich than all other FMs, FM3 was significantly richer than all other FMs, and there was no difference between FM2 and FM4 in richness. To assess the possible effects on the FM of shipping, habitation in a different institution, and the various phenotyping tests, representative samples were collected from the parent colonies used to generate the mice that were shipped to the Mouse Metabolic Phenotyping Center at the University of California-Davis and used in the phenotyping tests (Fig. 1b). Notably, two-factor ANOVA detected significant effects of both FM ($p < 0.001$, $F = 48.5$) and sex ($p < 0.001$, $F = 14.7$) on microbial richness in these samples, as well as a significant interaction between the two factors ($p = 0.005$, $F = 4.7$). FM3 and FM2 both demonstrated sex-dependent differences ($p < 0.001$, $t = 4.4$ and $p = 0.015$, $t = 2.5$, respectively), while no difference was detected between male and female mice colonized with FM1 or FM4. The significant pairwise differences between FMs were identical to those identified in the subset of mice used in phenotyping tests, i.e., FM1 < FM2 and FM4 < FM3 in terms of richness.

Subjective assessment of taxonomic composition at the level of genus confirmed previously observed FM-dependent differences in the relative abundance of dominant genera (Figs. 1c, S2). As the presence or absence of less abundant taxa can also influence microbial ecology and host responses, principal coordinate analysis (PCoA) plots were generated using unweighted Jaccard similarities, to demonstrate the significant differences in β-diversity between colonies, and similarity between the phenotyped mice and parent colonies (Fig. 1d). Testing via two-factor permutational multivariate ANOVA (PERMANOVA) validated this interpretation with significant FM-dependent ($p = 0.0001$, $F = 54.8$) and sex-dependent ($p = 0.0001$, $F = 8.4$) effects, as well as a significant interaction ($p = 0.0001$, $F = 5.9$). Pairwise comparisons suggest that the significant interaction reflects the pronounced sex-dependent effect observed in mice harboring FM3 (Table 1). A Venn diagram shows the number of ASVs detected in each FM profile (Fig. 1e), and comprehensive results of all FM- and sex-dependent differences in ASV relative abundance are provided in Fig. S3 and Supplementary Data 1.

**Supplier-origin FMs are associated with different locomotor activity, exploratory behavior, and anxiety-related behavior.** To assess differences in locomotor activity and exploratory behavior, open field testing was performed at 8 weeks of age. Open-field testing revealed several significant FM-associated differences, consistently indicating increased locomotor activity and exploratory behavior in mice colonized with FM4. For each mouse, data were collected for 20 min and stratified for analysis by 5-min intervals as period 1 (P1), P2, P3, and P4 allowing comparison of parameters during the entire test period or the final period following extended time in the test chamber. Based on the entire dataset, there was a significant FM-mediated effect on total distance traveled (Fig. 2a) and center distance traveled (Fig. 2b). Separate comparisons of distance traveled in the center

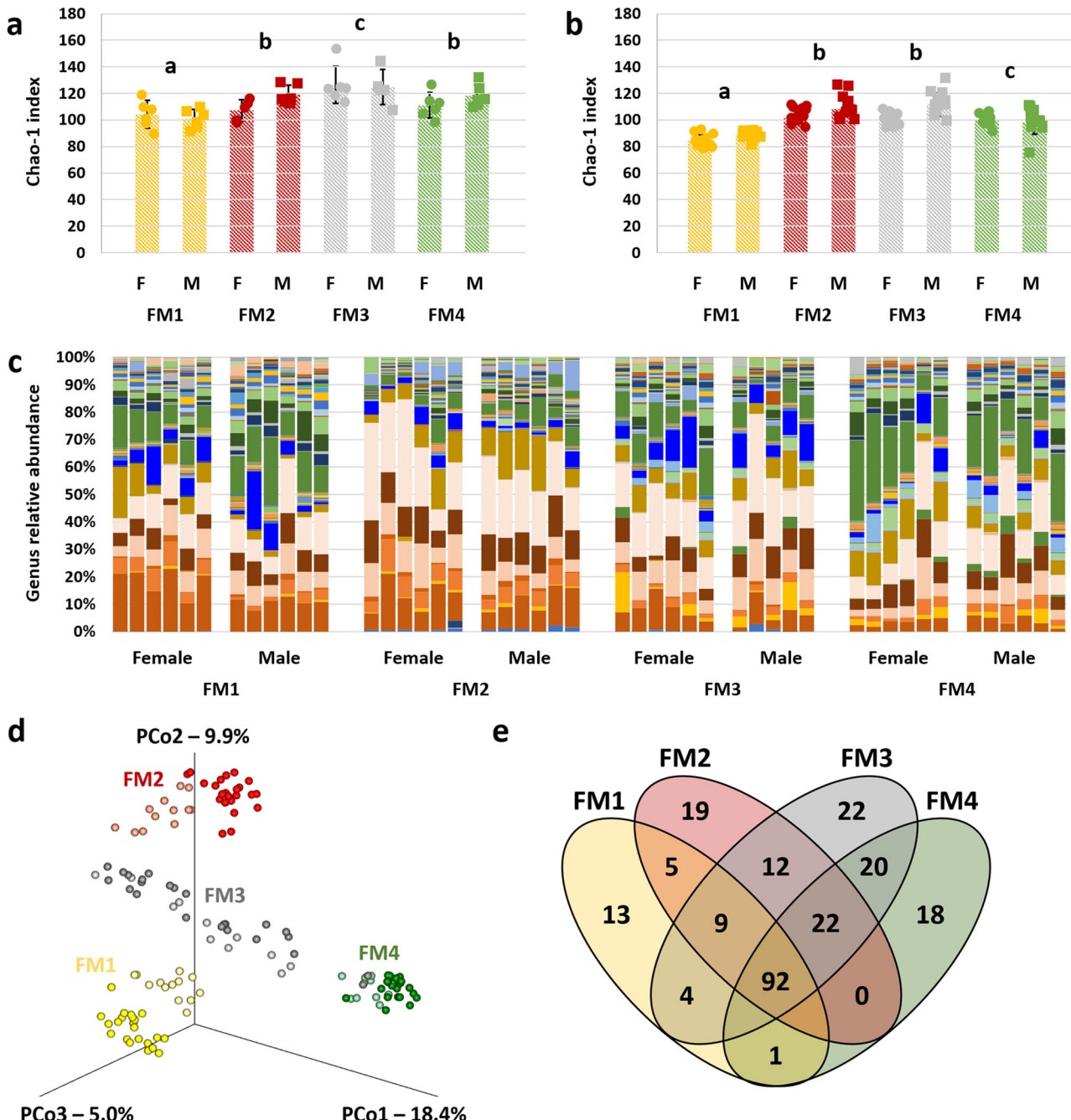

**Fig. 1 Colonies of CD-1 mice are colonized with four distinct fecal microbiomes. a** Dot plot showing the difference in microbial fecal richness ($p < 0.001$, $F = 10.1$), as represented by the Chao-1 index, in 16-week-old, female (F) and male (M) CD-1 mice colonized with fecal microbiome 1 (FM1), FM2, FM3, or FM4 ($n = 5$–6/sex/FM), and used in phenotyping tests. **b** Dot plot showing the difference in microbial fecal richness ($p < 0.001$, F = 48.5), as represented by the Chao-1 index, in 6 to 12-week-old, female (F) and male (M) CD-1 mice colonized with FM1, FM2, FM3, or FM4 ($n = 12$/sex/FM), collected from the parent colonies prior to shipping. **c** Stacked bar chart showing taxonomic diversity, at the level of genus, in female and male CD-1 mice ($n = 5$-6/sex/FM) harboring one of four supplier-origin gut microbiomes (FM1 through FM4). **d** Three-dimensional principal coordinate analysis (PCoA) plot, based on Jaccard similarities, showing β-diversity captured by the first three coordinates, of phenotyped mice (darker symbols, $n = 5$-6/sex/FM) and parent colonies (lighter symbols, $n = 12$/sex/FM). **e** Venn diagram showing the number of ASVs detected in at least one mouse of the various FM groups. $p$ and F values represent the main effects of FM in two-way ANOVA, and post hoc comparisons via Holm–Sidak method; significant pairwise FM-dependent differences are denoted by different letters, underlying bar charts represent mean ± SE.

and periphery during P4 also revealed significant FM-dependent effects with increased distance traveled in both portions of the chamber by mice colonized with FM4 relative to FM1 (Fig. 2c, d). There were also FM-dependent differences in the number of center entries (Fig. 2e), and in percent, time spent in the center (Fig. 2f), suggesting that these differences reflect greater

locomotor activity, as well as decreased thigmotaxis in mice with FM4. As an indicator of exploratory behavior, the number of rears (i.e., standing on hind legs) revealed similar patterns with a significantly greater number of rears across the entire time (Fig. 2g) and during P4 (Fig. 2h) in mice colonized with FM4, particularly in male mice. Sex-dependent effects were observed in

**Table 1 Results of pairwise comparisons of microbial richness, following two-factor ANOVA.**

| FM | Sex | Sex, within FM | | FM1, within sex | | FM2, within sex | | FM3, within sex | | FM4, within sex | |
|---|---|---|---|---|---|---|---|---|---|---|---|
| | | p | F | p | F | p | F | p | F | p | F |
| FM1 | Females | 0.019 | 2.42 | | | 0.0001 | 28.45 | 0.0001 | 28.83 | 0.0001 | 47.86 |
| | Males | 0.019 | 2.42 | | | 0.0001 | 27.44 | 0.0001 | 36.08 | 0.0001 | 43.17 |
| FM2 | Females | 0.0086 | 2.52 | 0.0001 | 28.45 | | | 0.0001 | 19.2 | 0.0001 | 41.22 |
| | Males | 0.0086 | 2.52 | 0.0001 | 27.44 | | | 0.0001 | 21.67 | 0.0001 | 29.59 |
| FM3 | Females | 0.0001 | 14.64 | 0.0001 | 28.83 | 0.0001 | 19.2 | | | 0.0001 | 39.85 |
| | Males | 0.0001 | 14.64 | 0.0001 | 36.08 | 0.0001 | 21.67 | | | 0.0001 | 13.29 |
| FM4 | Females | 0.0003 | 4.43 | 0.0001 | 47.86 | 0.0001 | 41.22 | 0.0001 | 39.85 | | |
| | Males | 0.0003 | 4.43 | 0.0001 | 43.17 | 0.0001 | 29.59 | 0.0001 | 13.29 | | |

Table showing p and F values associated with sex-associated differences in richness within each FM, and FM-associated differences in richness within each sex.

several of these and other parameters, with males often being more active relative to females (Table 2); however, differences during P4 in total distance traveled, peripheral distance traveled, and a number of center entries were purely associated with FM differences, with no significant sex-dependent effects or interactions. Collectively, these data indicate that mice of either sex colonized with FM4 were voluntarily more active, and exhibited more exploratory behavior.

Anxiety-related behavior was also assessed via automated analysis of time spent in a light/dark chamber. During a 10-min test period, a significant main effect of FM was detected in the percent time spent in the light portion of the box ($p < 0.001$, $F = 13.7$), and no significant effect of sex ($p = 0.60$) or sex × FM interaction ($p = 0.89$) was detected. Specifically, mice colonized with FM4 spent more time on the light side compared to all other groups ($p = 0.007$ vs. FM1, $p = 0.002$ vs. FM2, and $p < 0.001$ vs. FM3) suggesting reduced anxiety-related behavior in those mice (Fig. 3a). The number of transitions between light and dark sides of the chamber was also significantly different between FMs ($p = 0.02$, $F = 3.6$), with FM4 and FM2 being associated with the highest (85) and lowest (63) mean number of transitions, respectively (Fig. 3b), and no significant effect of sex ($p = 0.47$) or sex × FM interaction ($p = 0.24$). Supporting the observed difference in thigmotactic behavior, these results suggest anxiolytic effects of FM4 relative to other FMs.

Acoustic startle testing was performed to assess sensorimotor gating via prepulse inhibition (PPI) in mice with each FM. While no FM-mediated differences in PPI were detected, significant sex-mediated and FM-mediated differences were detected in the response to the initial acoustic startle ($p < 0.001$, $F = 14.7$ and $p = 0.006$, $F = 4.6$, respectively). Specifically, mice with FM2 were more responsive than mice with FM3 or FM4 (Fig. 3c). Auditory function, assessed via auditory brainstem response (ABR) tests of anesthetized mice, suggested impaired hearing in mice from all four groups, rendering the observed differences in acoustic startle difficult to interpret. Fear conditioning and fear testing were also performed and no differences were detected in cue-associated or context-associated fear conditioning.

**Supplier-origin FMs have significant effects on body morphology and physiology.** Throughout the phenotyping process, body weights were collected, allowing the generation of growth curves for each group of mice. While sex and age were both significantly associated with differences in weight ($p < 0.001$, $F = 657$ and $p < 0.001$, $F = 23.8$, respectively), FM was also a significant determinant ($p = 0.003$, $F = 42.6$). While no significant interactions were detected between a week and sex or FM, there was a significant FM × sex interaction ($p < 0.001$, $F = 5.8$), indicating FM-associated differences were not consistent across sexes.

Post hoc comparisons showed that mice with FM4, of both sexes, were smaller than their age- and sex-matched CD-1 counterparts with other FMs (Fig. 4a, b). In contrast, mice harboring FM2, which displayed relatively low activity levels and exploratory behavior, were substantially heavier than mice with other FMs. These differences in body weight were grossly apparent in lateral (Fig. S4) and dorsoventral (Fig. S5) radiographic images of the heaviest mice with FM4 and FM2, the two strains at either end of the spectrum. While there was not a significant FM-associated difference in body length, body length data correlated significantly with bodyweight at necropsy ($R^2 = 0.667$, $p = 2 \times 10^{-7}$) (Fig. S6), and the lack of a significant difference in body length was likely a function of the narrower range of values around the mean compared to body weight. The relatively smaller size of mice harboring FM4 was validated at necropsy via significant FM4-associated decrease in cardiac weight ($p = 0.003$, $F = 5.1$; Fig. 4c), a significant main effect of sex ($p < 0.001$, $F = 96.3$) and no significant interaction ($p = 0.55$). When normalized to body weight, no difference in cardiac weight was detected. ECG data collected at 12 weeks of age also revealed significant FM-dependent effects in mean SR amplitude ($p = 0.008$, $F = 4.3$; Fig. 4d) and duration of QRS interval ($p < 0.001$, $F = 6.9$; Fig. 4e), with no significant main effects of sex, or FM × sex interaction, in either measurement. As these features are typically attributable, in health, to either dampening of depolarization-repolarization by fluid or fat, or reduced myocardium[35], these features were attributed to the lower cardiac weight in mice with FM4. No sex- or FM-associated differences were detected in heart rate, RR interval, PR interval, QT interval, or QTc dispersion.

To test for differences in glucose metabolism, intraperitoneal glucose tolerance tests (IPGTT) were performed at 13 weeks of age, following a 12 h fast. Following IP injection of glucose (2 g/kg body weight), blood glucose was measured at 15, 30, 60, and 120 min post-injection, as the glucose was gradually taken up by peritoneal adipose tissue[36]. Results of a three-factor ANOVA confirmed time-dependent changes in blood glucose ($p < 0.001$, $F = 98.9$), a strong sexual dimorphism ($p < 0.001$, $F = 130.0$), and a significant difference between FMs ($p = 0.005$, $F = 4.4$) (Fig. 5a). Mice with FM4 had the highest overall area under the curve, while the larger, less active mice harboring FM2 had significantly faster glucose uptake than mice with the other FMs. No FM-associated difference was detected in plasma insulin levels at 16 weeks of age ($p = 0.21$, Fig. 5b), although a clear sex-dependent difference was detected ($p < 0.001$, $F = 21.4$).

**Supplier-origin FMs are associated with differences in lymphopoiesis.** Complete blood counts (CBCs) performed at necropsy also revealed significant FM-associated differences in the total white blood cell (WBC) count ($p = 0.016$, $F = 3.8$; Fig. 6a) with no significant effect of sex or interaction. Post hoc

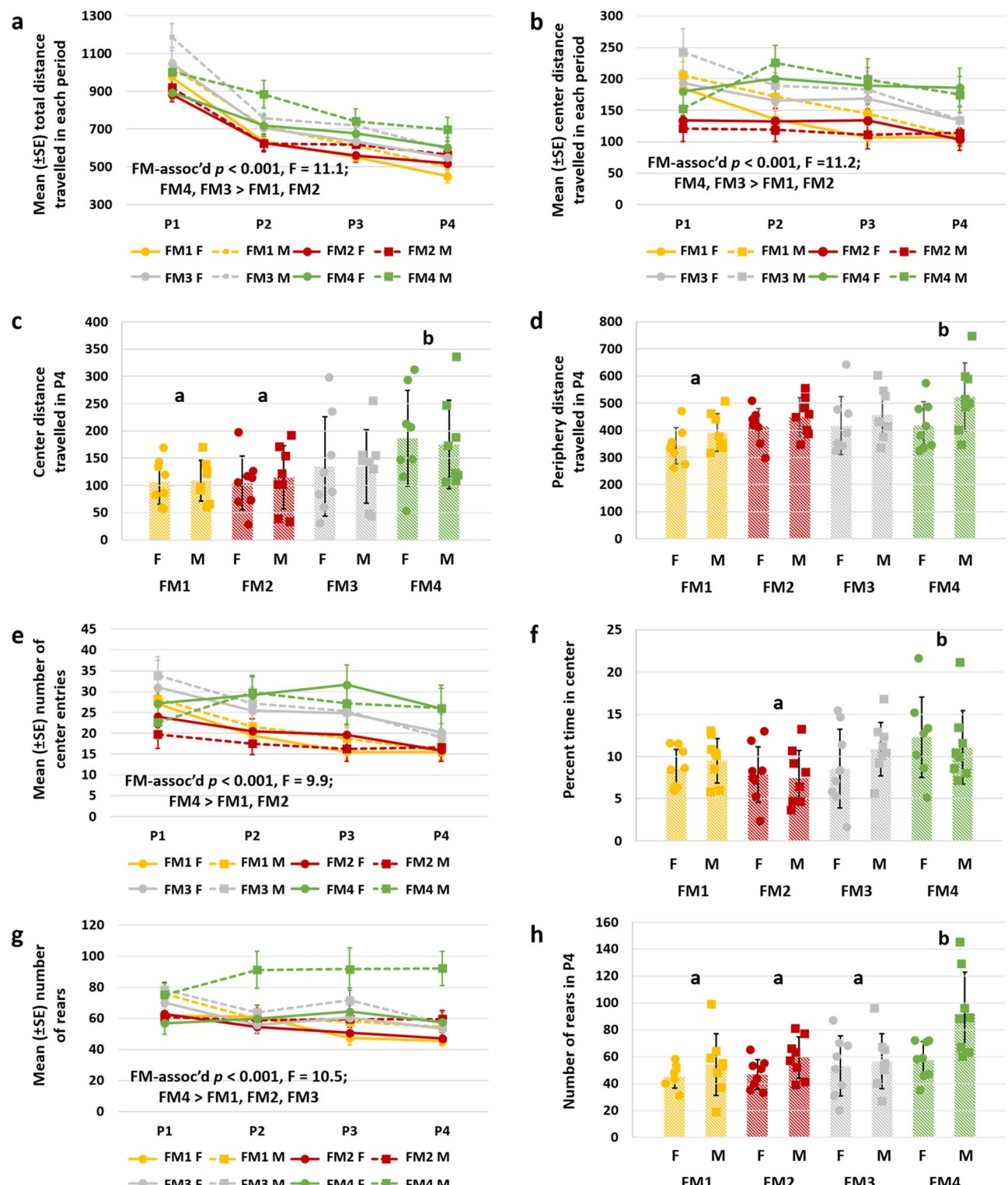

**Fig. 2 Supplier-dependent FMs are associated with the different locomotor and exploratory activity. a** Line chart showing mean (±SE) total distance traveled within each 5-min period (P1–P4) in the open field test. **b** Line chart showing mean (±SE) center distance traveled within each 5-min period (P1–P4). **c** Dot plot showing center distance traveled during the final 5-min period (P4), $p = 0.011$, $F = 4.1$. **d** Dot plot showing periphery distance traveled in P4 by the same mice. **e** Line chart showing the number of center entries during the entire time, $p = 0.014$, $F = 3.8$. **f** dot plot showing percent time in the center during the entire test period, $p = 0.026$, $F = 3.3$. **g** Line chart and **h** dot plot showing number of rears during the entire time and in P4 ($p = 0.003$, $F = 5.5$), respectively. All data generated in female (F) and male (M) CD-1 mice harboring FM1, FM2, FM3, or FM4 ($n = 8$/sex/FM). $p$ and $F$ values represent the main effects of FM in two-way ANOVA, and post hoc comparisons via Holm–Sidak method; significant pairwise FM-dependent differences are denoted by different letters, $n = 8$/sex/FM at all time-points, underlying bar charts represent mean ± SE.

**Table 2 Results of two-factor ANOVA of behavioral parameters during P4.**

| Outcome measure | Sex (main effect) | | Sex (post hoc) | FM (main effect) | | FM (post hoc) | Interaction | |
|---|---|---|---|---|---|---|---|---|
| | p | F | | p | F | | p | F |
| Center distance traveled | 0.98 | 0.0004 | | 0.011 | 4.1 | FM4 > FM1, FM2 | 0.98 | |
| Peripheral distance traveled | 0.011 | 6.9 | M > F | 0.014 | 3.8 | FM4 > FM1 | 0.65 | |
| Percent time center | 0.66 | 0.2 | | 0.026 | 3.3 | FM4 > FM2 | 0.55 | |
| Total distance traveled | 0.096 | 2.9 | | 0.006 | 4.5 | FM4 > FM1 | 0.94 | |
| Number of rears | 0.003 | 9.3 | M > F | 0.003 | 5.4 | FM4 > all others | 0.13 | |
| Number of center entries | 0.990 | 0.0002 | | 0.023 | 3.4 | FM4 > FM1, FM2 | 0.99 | |

Table showing p and F values associated with main effects of Sex and FM, the directionality of post hoc comparisons between male (M) and female (F) and the four fecal microbiotas (FM1–FM4), and interactions.

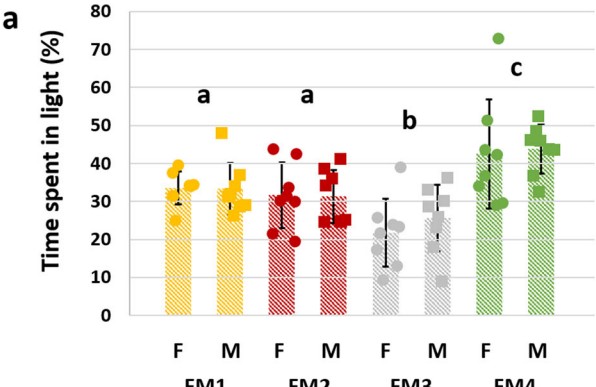
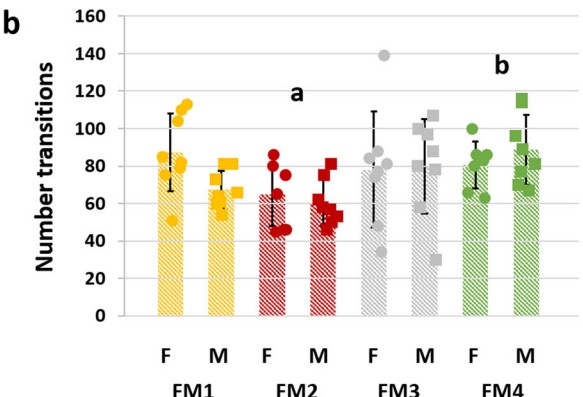
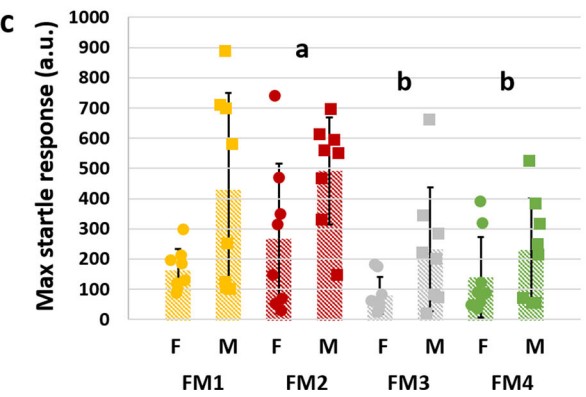

**Fig. 3 Supplier-dependent FMs are associated with different anxiety-related behavior. a** Dot plot showing the percent time spent in the illuminated half of a light/dark chamber by female (F) and male (M) mice colonized with FM1, FM2, FM3, or FM4 ($p < 0.001$, $F = 13.7$). **b** Dot plot showing the total number of transitions between the two halves of a light/dark chamber by female and male mice colonized with FM1, FM2, FM3, or FM4 ($p = 0.02$, $F = 3.6$). **c** Dot plot showing the response to startle in arbitrary units (a.u.) by female and male mice colonized with FM1, FM2, FM3, or FM4 ($p = 0.006$, $F = 4.6$). p and F values represent main effects of FM in two-way ANOVA, and post hoc comparisons via Holm–Sidak method; significant pairwise FM-dependent differences are denoted by different letters, $n = 8$/sex/FM at all time-points, underlying bar charts represent mean ± SE.

comparisons found significant differences in total WBC count between mice colonized with FM2 and FM4. The greater number of leukocytes in mice with FM4 was due to an absolute lymphocytosis as total lymphocyte numbers mirrored the differences in total WBC ($p = 0.007$, $F = 4.6$, Fig. 6b). While no other FM-dependent differences in total erythrocyte (RBC), platelet, neutrophil, monocyte, eosinophil, or basophil number were detected, the relative proportion of lymphocytes and neutrophils paralleled the total cell counts and indicated that the differences in frequency of lymphocytes between different FMs ($p = 0.012$, $F = 4.0$, Fig. 6c) was offset by differences in the frequency of neutrophils

($p = 0.006$, $F = 4.7$, Fig. 6d). The sex-dependent difference in the CBC was limited to differences in platelet count and, unexpectedly, a greater total number ($p = 0.002$, $F = 10.7$) and percentage ($p = 0.009$, $F = 7.4$) of monocytes in peripheral blood from females.

**No FM-dependent differences in grip strength or ophthalmological parameters were detected.** Several additional phenotyping assays were performed for which no significant FM-associated differences were detected. These included evaluation with the

modified SmithKline Beecham, Harwell, Imperial College, Royal London Hospital, phenotype assessment (SHIRPA) protocol[37,38] and measurement of grip strength at 8 weeks of age, and complete ophthalmic examination at 15 weeks of age. All source data are available as Supplementary Data 2, and histologic slides of multiple organs are available upon reasonable request.

## Discussion

Collectively, these data generated in healthy adult mice from the same background stock and maintained as separate, minimally inbred colonies, provide strong evidence for FM-mediated effects on baseline locomotor and exploratory activity. The observed differences in overall activity and anxiety-related behavior (e.g., reduced thigmotaxis, greater time spent in the light portion of a light/dark chamber), were associated with multiple differences in downstream physiological parameters such as overall body size, heart weight and contractility, and glucose metabolism. While we and others have previously demonstrated significant effects of these different FM profiles on susceptibility to various disease models[39–44], these differences in behavior and basic physiological parameters have wide-reaching implications in biomedical research reliant on mouse models.

These findings bring into question the interpretation of previous studies documenting behavioral and physiological differences between substrains from different suppliers[7–28]. While genetic differences are surely responsible for a portion of the reported substrain differences, the FM of mice in such studies is seldom controlled or normalized and the current data suggest that supplier-dependent differences in the FM may also contribute to phenotypic differences in behavior and basic physiology. Moreover, data generated in F1 offspring interpreted as evidence of an X-linked substrain characteristic[15], and F2 offspring failing to link candidate substrain polymorphisms[27], could both be explained by the transmission of phenotype-associated maternal FM to offspring. The phenotypic differences reported here are now subject to further fully crossed studies designed to partition the influence of host genotype and FM on the phenotype of interest, and identify interactions between the two. One attractive candidate would be the host genomic and metagenomic features associated with the increased voluntary locomotor activity. While multiple studies[13,14] have reported increased activity in C57BL/6J (B6J) mice compared to C57BL/6N (B6N) mice, in apparent contrast to our study wherein FM1 (derived from B6J mice) was associated with decreased locomotor activity relative to FM4 (derived from B6N). This apparent interaction begs further investigation.

Multiple studies have shown an anxiolytic phenotype in GF mice, and no detectable difference between GF and SPF mice in locomotor activity[32,45,46]. In the studies showing the transfer of behavioral phenotypes between inbred SPF BALB/c and outbred SPF Swiss Webster mice via reconstitution of GF mice with anxiogenic or anxiolytic FM profiles from the reciprocal strain of SPF donor[33], it is possible that host-associated genetic factors have influenced the FM to assume certain anxiolytic or anxiogenic properties. The fact that FM transfer can also confer behavioral properties of the donor demonstrates the influence of the FM on behavior, but is nonetheless confounded by the potential for a circular relationship wherein host factors somehow select for the associated FM. The current work expands on those studies by controlling for the effects of host genetics while avoiding the use of artificial systems such as GF and antibiotic-treated mice. Rather, our findings show convincingly that different, naturally occurring FM profiles acquired at birth affect baseline locomotor and exploratory activity in healthy outbred mice, and substantiate a role for the FM in anecdotal reports of different behavioral phenotypes in substrains of mice from different suppliers.

The hematological differences are of interest in the context of several recent studies convincingly showing differential disease severity in models of infectious disease associated with different supplier-origin microbiota. While some studies have suggested that $T_H17$-mediated immunity induced by segmented filamentous bacteria (SFB), a taxon lacking from the Jackson FM but present in many other supplies, may be responsible for colonization against pathogens (e.g., *Citrobacter rodentium*)[44], it is now clear that other taxa and mechanisms likely contribute to FM-dependent differences in colonization resistance against *Enterobacteriaceae* including intra-family competition[47–50]. Recent work from Patrick Schloss' lab investigating features within different lab and vendor-origin FMs associated with clearance of *Clostridioides difficile* underscores that the effects of these different FMs may be associated with more than one single taxon[51]. Similarly, multiple studies have revealed significant vendor FM-dependent differences in parasitemia and mortality following infection with *Plasmodium yoelli* in mouse models of malaria. Of note, while these studies largely agree that Jackson or Taconic-origin FM are protective when compared to the richer FM of CRL or Envigo[41,52,53], these findings are in contrast to the aforementioned greater susceptibility to *Salmonella* or *Citrobacter* infection associated with the Jackson FM. Taken together, these findings suggest that each FM has selective effects on host immunity or physiology resulting in either increased or decreased susceptibility to infectious disease, depending on the pathogen in question and mechanisms by which the pathogen is cleared. Our data indicate that the four vendor-origin FMs are associated with inherent differences in lymphopoiesis, with enhanced lymphocyte production associated with FM4, resulting in an absolute leukocytosis relative to the other FMs. The clinical significance of this is unclear but noteworthy in terms of experimental reproducibility.

Recovery of cryopreserved germplasm via ET has traditionally used outbred CD-1 mice as surrogate dams due to their favorable breeding indices and maternal performance, and these CD-1 mice function as FM donors when used as surrogate dams. Several studies have demonstrated significant FM-mediated effects on disease model phenotypes, highlighting the utility of these approaches in optimizing mouse models, as well as identifying microbial taxa associated with disease and potentially translatable to the human condition. Here we demonstrate, in a controlled fashion and using a robust study design and testing platform, that these colonies can also be used to investigate host/FM interactions in neurodevelopment and behavior, body morphology, glucose metabolism, and cardiac physiology.

## Methods

**Mice.** All activities described here were performed in accordance with the guidelines put forth in the Guide for the Care and Use of Laboratory Animals, and were approved by the Institutional Animal Care and Use Committee (IACUC) of the University of Missouri (protocol 9737) and the IACUC of the University of California-Davis (protocol 20863). All mice (*Mus musculus*) tested or sampled in the current study were outbred Crl:CD1(ICR) (i.e., CD-1) mice (Charles River Laboratory), generated in four separate, minimally inbred colonies. Colonies of mice initiated at Discovery Ridge, in Columbia, MO, were housed under barrier conditions in microisolator cages with compressed pelleted paper bedding and nestlets, on ventilated racks (Thoren, Hazleton, PA) with ad libitum access to irradiated chow (LabDiet 5058, LabDiet, St. Louis, MO) and acidified, autoclaved water, under a 14:10 light/dark cycle. Water was acidified using an automated bottle filler (model 9WEF, Tecniplast, Buguggiate, Italy) designed to titrate municipal water with sulfuric acid to a target pH of 2.5 (range 2.3–2.7). Mice were determined to be free of all overt and opportunist bacterial pathogens including *Bordetella bronchiseptica*, cilia-associated respiratory (CAR) bacillus, *Citrobacter rodentium*, *Clostridium piliforme*, *Corynebacterium bovis*, *Corynebacterium kutscheri*, *Helicobacter* spp., *Mycoplasma* spp., *Pasteurella pneumotropica*, *Pneumocystis carinii*, *Salmonella* spp., *Streptobacillus moniliformis*, *Streptococcus pneumoniae*; adventitious viruses including H1, Hantaan, KRV, LCMV, MAD1, MNV, PVM, RCV/SDAV, REO3, RMV, RPV, RTV, and Sendai viruses; intestinal protozoa including *Spironucleus muris*, *Giardia muris*, *Entamoeba muris*, trichomonads, and other large intestinal flagellates and amoebae; intestinal parasites

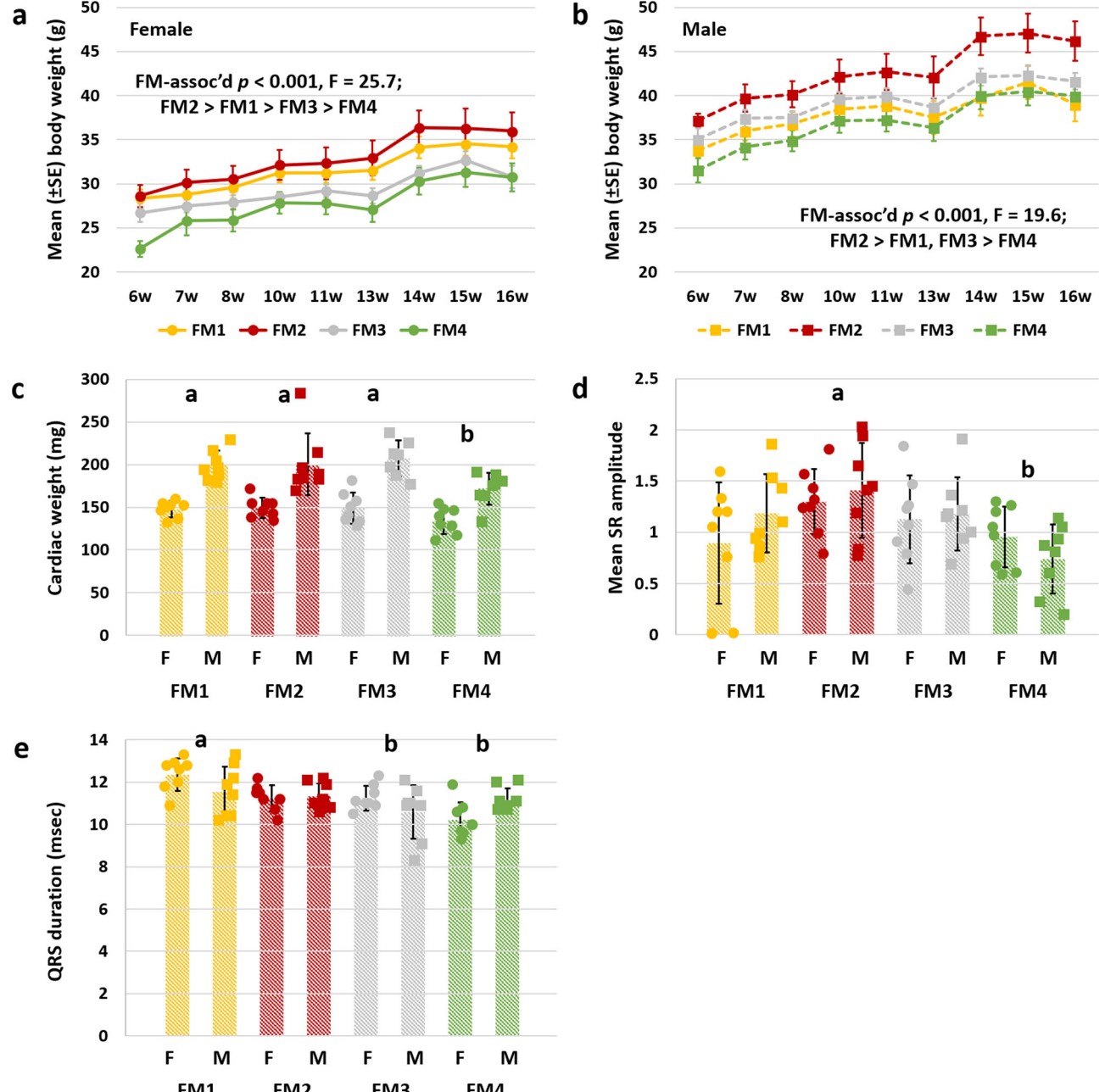

**Fig. 4 Supplier-dependent FMs are associated with differences in body size and cardiac force. a** Line chart showing mean (±SE) body weight of female CD-1 mice colonized with FM1, FM2, FM3, or FM4 (legend at bottom), from 6 weeks (6w) to 16 weeks of age. **b** Line chart showing mean (±SE) body weight of male CD-1 mice colonized with FM1, FM2, FM3, or FM4 (legend at bottom), from 6 weeks to 16 weeks of age. **c** Dot plot showing cardiac weight at 16 weeks, in female (F) and male (M) mice colonized with FM1, FM2, FM3, or FM4 ($p = 0.003$, $F = 5.1$). **d** Dot plot showing SR amplitude, measured via ECG, in mice shown in Panels A-C ($p = 0.008$, $F = 4.3$). **e** Dot plot showing QRS interval, measured via ECG, in mice shown in Panels **a–c** ($p < 0.001$, $F = 6.9$). $p$ and $F$ values represent main effects of FM in two-way ANOVA, and post hoc comparisons via Holm–Sidak method; significant FM-dependent differences are denoted by different letters, $n = 8$/sex/FM at all time-points, underlying bar charts represent mean ± SE.

including pinworms and tapeworms; and external parasites including all species of lice and mites, via quarterly sentinel testing performed by IDEXX BioAnalytics (Columbia, MO). Mice used in all phenotyping assays were shipped to UC Davis at 3 weeks of age. While at UC Davis, mice were socially housed with other mice of the same FM in individually ventilated cages (Optimice IVC, Animal Care Systems, Centennial, CO) on a 12:12 h light cycle, and received standard laboratory rodent chow (Harlan Teklad diet 2918) and autoclaved water.

FM samples were collected from phenotyped mice at 16 weeks of age ($n = 5$–6 per sex per FM), frozen immediately, and shipped to the MUMC for processing. Additional fecal samples were collected from the parent colonies used to generate the mice used in phenotyping assays. These samples, collected in mid-2018, comprised 24 adult mice/FM ($n = 12$ female and 12 male/FM). Fecal samples were collected according to previously published methods[54]. Briefly, mice were placed in

an empty autoclaved housing cage, within a laminar flow hood, and allowed to defecate naturally. Freshly evacuated fecal pellets were then immediately collected into a sterile collection tube using autoclaved wooden toothpicks, which were discarded after an individual use. Sample tubes were kept on ice while collecting from all mice and then transported to a −80 °C freezer until DNA extraction was performed.

**Fecal DNA extraction**. Fecal DNA was extracted using PowerFecal DNA Extraction kits (Qiagen), according to the manufacturer's instructions, with the exception that the initial disaggregation of samples was performed in the supplied bead tubes using a TissueLyser II (Qiagen) at 1/30 Hz for 3 min in lysis buffer.

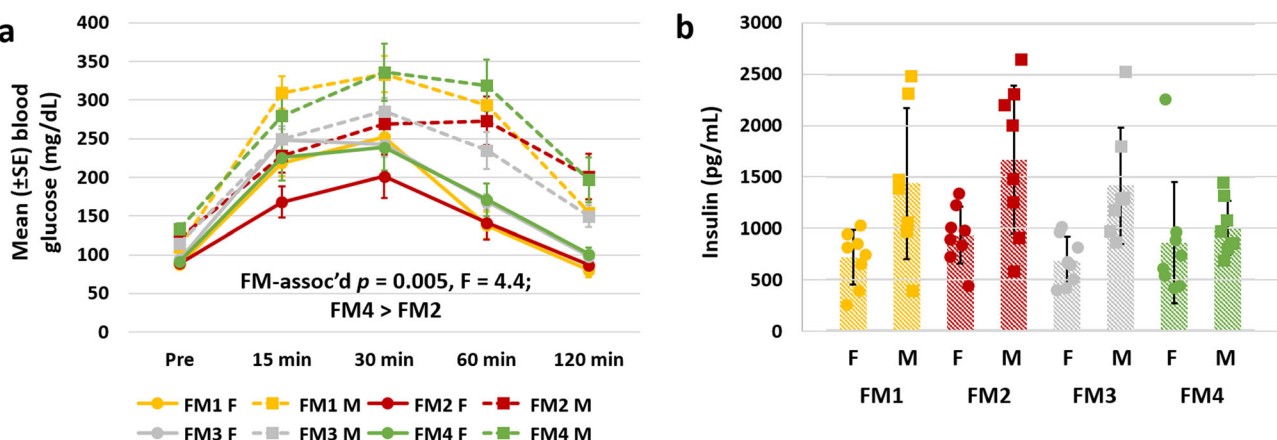

**Fig. 5 Supplier-dependent FMs are associated with differences in glucose metabolism. a** Line chart showing mean (±SE) blood glucose (mg/dL) at baseline (Pre), 15, 30, 60, and 120 min post-IP glucose injection during an intra-peritoneal glucose tolerance test performed at 13 weeks of age in female (circles, solid lines) and male (squares, dotted lines) CD-1 mice colonized with one of four supplier-origin gut microbiomes (legend at bottom). **b** dot plot showing sex-dependent difference ($p < 0.001$, $F = 21.4$) in serum insulin levels (pg/mL) at 16 weeks of age in female (F) and male (M) mice, $n = 8$/sex/FM at all time-points, underlying bar charts represent mean ± SE.

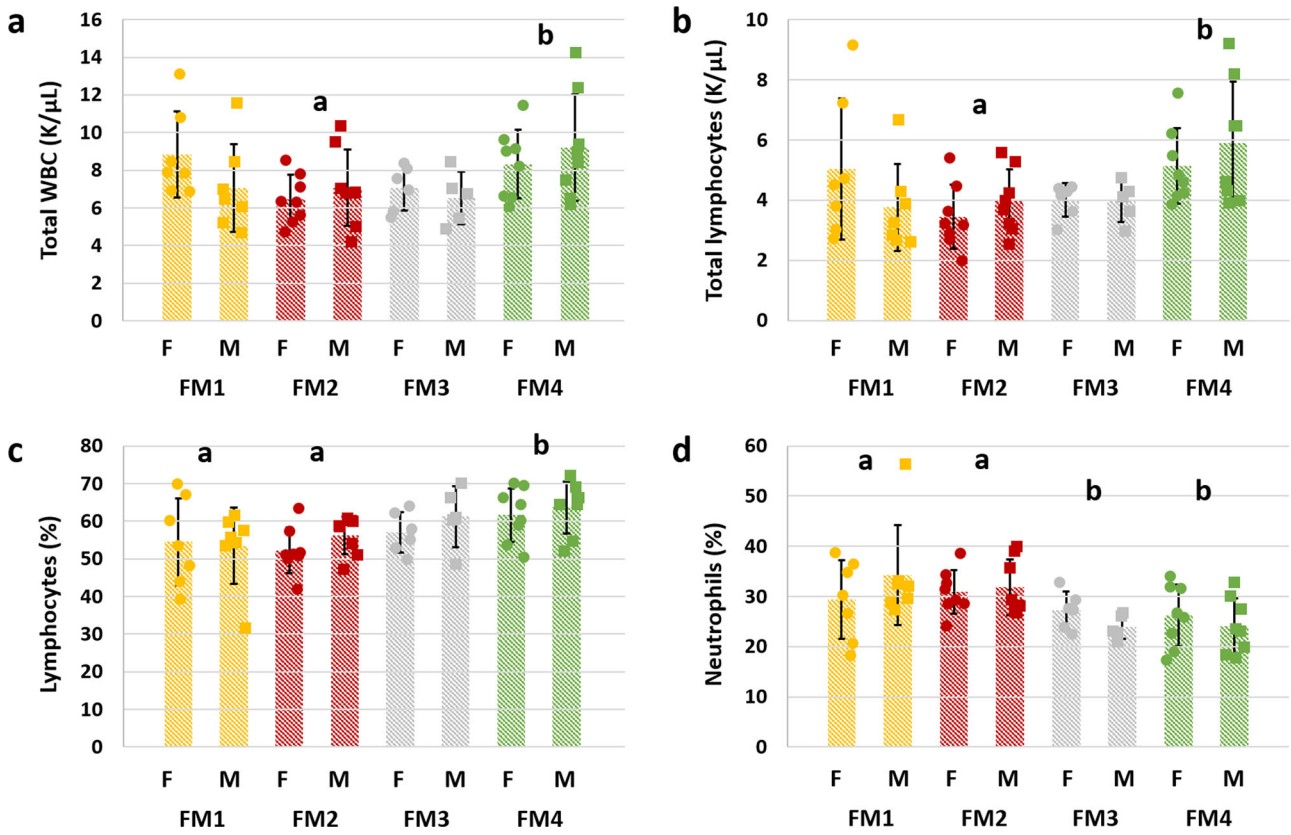

**Fig. 6 Supplier-dependent FMs are associated with differences in peripheral leukocyte populations.** Dot plots showing **a** total white blood cell (WBC) count (K/μL) ($p = 0.016$, $F = 3.77$), **b** total lymphocyte count (K/μL) ($p = 0.007$, $F = 4.6$), **c** percent lymphocytes ($p = 0.012$, $F = 4.0$), and **d** percent neutrophils ($p = 0.006$, $F = 4.7$) in peripheral blood of female (F) and male (M) CD-1 mice colonized with one of four supplier-origin gut microbiomes (legend at bottom). $p$ and $F$ values represent the main effects of FM in two-way ANOVA, and post hoc comparisons via Holm–Sidak method; significant FM-dependent differences are denoted by different letters, $n = 8$/sex/FM at all time-points, underlying bar charts represent mean ± SE.

DNA was eluted in 100 μL buffer and quantified using Quanti-iT BR dsDNA kits and Qubit 2.0 fluorometer (Thermo Fisher).

**16S rRNA library preparation and sequencing**. Extracted fecal DNA was processed at the University of Missouri DNA Core Facility. Bacterial 16S rRNA amplicons were constructed via amplification of the V4 region of the 16S rRNA gene with universal primers (U515F/806R) previously developed against the V4

region, flanked by Illumina standard adapter sequences[55,56]. Oligonucleotide sequences are available at proBase[57]. Dual-indexed forward and reverse primers were used in all reactions. PCR was performed in 50 μL reactions containing 100 ng metagenomic DNA, primers (0.2 μM each), dNTPs (200 μM each), and Phusion high-fidelity DNA polymerase (1U). Amplification parameters were 98 °C$^{(3:00)}$ + [98 °C$^{(0:15)}$ + 50 °C$^{(0:30)}$ + 72 °C$^{(0:30)}$]×25 cycles +72 °C$^{(7:00)}$. Amplicon pools (5 μL/reaction) were combined, thoroughly mixed, and then purified by addition of Axygen Axyprep MagPCR clean-up beads to an equal volume of 50 μL of

amplicons and incubated for 15 min at room temperature. Products were then washed multiple times with 80% ethanol and the dried pellet was resuspended in 32.5 μL EB buffer, incubated for 2 min at room temperature, and then placed on the magnetic stand for 5 min. The final amplicon pool was evaluated using the Advanced Analytical Fragment Analyzer automated electrophoresis system, quantified using quant-iT HS dsDNA reagent kits, and diluted according to Illumina's standard protocol for sequencing on the MiSeq instrument.

**Bioinformatics**. The DNA sequences were assembled and annotated at the MU Informatics Research Core Facility. Primers were designed to match the 5′ ends of the forward and reverse reads. Cutadapt[58] (version 2.6; https://github.com/marcelm/cutadapt) was used to remove the primer from the 5′ end of the forward read. If found, the reverse complement of the primer to the reverse read was then removed from the forward read as were all bases downstream. Thus, a forward read could be trimmed at both ends of the insert was shorter than the amplicon length. The same approach was used on the reverse read, but with the primers in the opposite roles. Read pairs were rejected if one reads or the other did not match a 5′ primer, and an error rate of 0.1 was allowed. Two passes were made over each reading to ensure the removal of the second primer. A minimal overlap of 3 with the 3′ end of the primer sequence was required for removal.

The QIIME2[59] DADA2[60] plugin (version 1.10.0) was used to denoise, de-replicate, and count ASVs (amplicon sequence variants), incorporating the following parameters: (1) forward and reverse reads were truncated to 150 bases, (2) forward and reverse reads with a number of expected errors higher than 2.0 were discarded, and (3) Chimeras were detected using the consensus method and removed. R version 3.5.1 and Biom version 2.1.7 were used in QIIME2. Taxonomies were assigned to final sequences using the Silva.v132 database[61], using the classify-sklearn procedure.

**Phenotyping**. All phenotyping assays were performed at the MMPC at UC Davis, according to their standardized and well-validated protocols between 0800 and 1600 Pacific Standard Time. CD-1 mice harboring each FM were bred at the MU MMRRC, and shipped to the MMPC within one week of weaning at exactly 3 weeks of age ($n = 8$ per sex per FM). Mice of different FMs were shipped in separate shipping containers, and maintained on different shelves of the IVC (under positive pressure) at the MMPC, to avoid cross-contamination and contamination with other microbes. As the same mice were subjected to multiple testing platforms, tests were performed at intervals between weeks 5 and 16 of age (Fig. S1). Body weights were collected at 6, 7, 8, 10, 11, 13, 14, 15, and 16 weeks of age, by placing the mouse in a sterile container on a tared scale. Where possible, the uniform protocols of the International Mouse Phenotyping Consortium (IMPC) used in the phenotyping tests are included as links below, to provide the reader additional details.

**Grip strength**. The grip strength test (ESLIM_009_001) measures forelimb and hindlimb muscle strength as an indicator of neuromuscular function. For forelimb grip strength, mice were held by the base of their tails so only the front paws grip the platform and pulled back steadily until the grasp was released. Grip strength for both sets of limbs was performed similarly with both forepaws and hind paws to attach to the grid. The grip strength sensor measures the maximum force applied (in grams) once the grasp is released (Chatillion, model E-DFE-002, Columbus Instruments). The mean of three consecutive trials for forelimbs and the combined forelimb/hind limb were taken.

**Open-field testing**. The open-field test (BCM_OFD_001) was used to evaluate locomotion and exploratory behaviors as well as behaviors associated with anxiety (i.e., thigmotaxis). After an acclimation period of 30 min, mice were placed at the periphery of a 44.5 × 44.5 × 20 (cm) clear Plexiglas arena with their head facing the wall and allowed to explore for 20 min. Movement and time spent in specific zones were tracked by infrared sensors and captured on video (Opto Varimex 4, Columbus Instruments, Columbus, OH). Parameters were collected over the total duration (single), or by 5 min increments (series) to evaluate habituation and learning.

**Modified SHIRPA**. The modified SmithKline Beecham, Harwell, Imperial College, Royal London Hospital phenotype assessment (SHIRPA) protocol (ALTESLIM_008_001) consists of a battery of comprehensive observational measures of behaviors, a first-line screening that scores more than 40 specific unprovoked behaviors and morphological parameters. The setup includes a cylindrical viewing jar, a larger arena with clearly defined squares, a wire grid, and a click box. The protocol provides an assessment of muscle, cerebellar, sensory, and autonomic functions as well as physical and morphological characteristics. Observations included posture, 10 min-locomotor activity, and gait quality, presence of tremors, transfer arousal, tail position, touch and startle responses, tail hang and righting responses, vocalizations, coat color, and quality; aggressive or abnormal behaviors were noted.

**Light/dark box testing**. The purpose of this procedure (IMPC_LDT_001) was to monitor the anxiety-related behavior of mice in a two-chambered light–dark arena over a defined period of time (20 min) in order to assess the anxiety phenotype. The test apparatus consists of a box divided into a dark chamber (4–7 lx) and a brightly illuminated chamber (100–200 lx). Mice were introduced into the light side of the arena, facing the wall. Movements of mice between the two chambers were tracked by infrared sensors and captured on video (Opto Varimex-4, Columbus Instruments, Columbus, OH). First latency to enter the dark compartment and total time spent in the dark compartment are indices of bright-space anxiety in mice.

**Acoustic startle and prepulse inhibition**. The startle reflex measures the acoustic function of mice but can also be modified by several host factors, while the prepulse inhibition measures the ability of mice to adapt to a strong sensory stimulus (startle) when a preceding weaker signal (prepulse) is given, to evaluate the sensory-gating capacity, or the ability to integrate sensory relevant information. In this protocol (IMPC_ACS_002), mice were placed in an acoustic chamber, in a tubular enclosure coupled to a motion sensor that records the startle response (SR-LAB startle response system, San Diego). After habituation, mice were exposed to short bursts of sound (40-ms 120-dB acoustic startle). A pressure gauge under the floor chamber measures the startle (jump) response. The eight stimuli (background (65 dB), three prepulses (74, 82 and 90 dB) presented alone or preceded by the startle, and a maximum startle (120 dB)) were repeated 10 times in a random order; the intertrial iteration was ~15 s. Software was used to calculate the attenuation of a startle (% prepulse inhibition) (SR-LAB Startle Response System, San Diego, CA).

**Fear conditioning and testing**. This procedure (ICS_FEA_001), consisting of two 10-minute sessions conducted over two days, was used to measure aversive learning and memory. After acclimation to the test chamber, mice were submitted to the following conditioning program (1) 120 s baseline period; (2) 30 s of neutral conditioned stimulus (cue: 2800 Hz, 85 dB tone), paired with a two-second aversive unconditioned stimulus (mild [0.75 mA] foot shock); (3) 150 s of lag time. During this test, mice also associate the background context with the conditioned stimulus. The next day, mice were introduced to the same environment for 300 s without stimuli (to test contextual memory), and then placed in a new environment with tone (to test cued memory). Freeze time and locomotor activity were captured by a video-tracking system (NIR Video Fear Conditioning System, Med Associates Inc, St-Albans, VT).

**Auditory brainstem response**. The purpose of this test (IMPC_ABR_002) was to measure the hearing sensitivity and physiologic parameters associated with hearing, using evoked recordings in anesthetized mice. Mice were anesthetized using a mixture of ketamine (100 mg/kg) and xylazine (10 mg/kg). Sterile ophthalmic lube was applied to each eye. Once immobile by toe pinch assessment and abolishment of the righting reflex, subdermal needle electrodes were placed on the vertex and overlying each bulla. The anesthetized mouse was placed inside an acoustic chamber in the ventral position. A test trace was recorded as a system control at 70 dB and then a series of ABR were recorded from 0 to 95 dB in 5 dB steps. The entire testing procedure takes approximately 30 min per mouse. Mice were returned to their home cage located on a heating pad and monitored according to campus guidelines until recovery was complete and they could be returned to the vivarium.

**Non-invasive cardiac electrophysiology**. Electrocardiogram (ECG) data were captured noninvasively from conscious mice (IMPC_ECG_002). After a 30-min acclimation period, each mouse placed on an electrode-fitted platform and further acclimated for 10 min prior to the initiation of data collection. Once the mouse was acclimated to the platform, it was placed in an acrylic housing chamber containing three gel-coated footpad wells. The cardiac signals were recorded and analyzed using ECGenie Instrument (eMouse Specifics Inc, Framingham, MA).

**Intra-peritoneal glucose tolerance test (IPGTT)**. The glucose tolerance test (IMPC_IPG_001) was used to measure the clearance of intraperitoneally injected glucose via adipose tissue. Mice were fasted for ~18 h prior to the procedure. Control blood glucose blood samples were taken by alcohol cleaning and then scoring the tip of the tail with a sterile scalpel blade and milking a small (5 μL) sample onto a glucose test strip. The bleeding was controlled through the application of dry gauze and pressure hemostasis. Following the initial blood sampling, 20% glucose solution was injected intraperitoneally at a dose of 2 g/kg (μL volume injected = 10 × body weight in grams; average mouse 0.25–0.3 mL). Blood glucose measurements were then taken at 15, 30, 60, and 120 min after glucose injection by agitation and removal of the previous tail tip clot and milking of ~5 μL of blood onto the test strip. Total blood collected for the procedure was 25 μL per mouse.

**Radiography**. Anesthetized mice were positioned and imaged in a Faxitron X-ray system (Tucson, AZ) (IMPC_XRY_001). Five different scans were taken of each mouse including the whole body (dorsoventral and lateral), head (dorsoventral and lateral), and whole arm from the elbow (dorsoventral). The entire process lasted

~15–20 min per mouse. Mice were returned to their home cage located on a heating pad and monitored according to campus guidelines until recovery was complete and they could be returned to the vivarium.

**Ophthalmic exam**. Two-minute eye exams were performed by a veterinarian resident (UC Davis Vet School), on conscious mice restrained by trained phenotyping staff. A slit lamp was used to assess the iris, cornea, and lens. Subsequently, a topical dilation solution was applied to both eyes (50:50 tropicamide/phenylephrine solution), for fundic exam of the posterior chamber of the eyes (retina, optic disc, and vitreous humor) using an ophthalmoscope.

**Necropsy and Hematology**. At 16 weeks of age, mice were humanely euthanized according to the guidelines set forth by the AVMA Statement on Humane Euthanasia, the Guide for the Care and Use of Laboratory Animals. Mice were then examined grossly according to the standardized protocols of the IMPC (HAS_PAT_002), and various data were collected in a uniform fashion including body weight and length, and cardiac weight (IMPC_HWT_001). Blood collected immediately post mortem via cardiocentesis was used for hematology using a clinical, flow cytometric hematology analyzer, and insulin measurement according to their respective IMPC protocols (ICSLA_HEM_002 and IMPC_INS_003).

**Statistics and reproducibility**. Two-way permutational multivariate analysis of variance (PERMANOVA) was used to test for significant main effects of sex and FM in β-diversity, followed by one-way PERMANOVA to provide pairwise comparisons between FM within sex, and between sex within FM. PERMANOVA testing was performed using PAST software and was based on Jaccard similarities. Univariate data were first tested for normality using the Shapiro–Wilk method, and then groups were compared using the appropriate parametric or nonparametric tests. In the case of single time-point comparisons (e.g., sex- and FM-associated differences in richness, total percent time spent in the center of the open field test, or cardiac weight at necropsy) two-way ANOVA and Kruskal–Wallis ANOVA on ranks was used for normally and non-normally distributed data, respectively. For comparisons including time-point as a third factor (e.g., total distance traveled per interval in the open field test, growth curve, or glucose levels during the IPGTT), three-way ANOVA was used and reported differences between FM in such situations reflect the main effects of FM. In all cases, post hoc comparisons were made using the Holm–Sidak method. Univariate statistical analyses were performed using SigmaPlot 14.0 (Systat Software, Inc, San Jose, CA).

**Reporting summary**. Further information on research design is available in the Nature Research Reporting Summary linked to this article.

## Data availability

All 16 S rRNA sequencing data have been deposited in the National Center for Biotechnology Information (NCBI), Sequence Read Archive (SRA), and are available as BioProject PRJNA683598. All other source data are available within the Supplementary Data files, or from the corresponding author upon request.

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

## Acknowledgements
We would like to acknowledge the staff of the MU DNA Core and Informatics Research Core facilities for their expertise in 16S rRNA library preparation and sequencing, and downstream bioinformatics, Giedre Turner and Becky Dorfmeyer for extraction and preparation of fecal DNA, Armedia O'Neill-Blair for her work in management of the CD-1 colonies, and Karen Clifford for assistance with figure formatting.

## Author contributions
ACE and CLF conceived and designed the experiment, analyzed and interpreted the data. KCKL, RA, LRB, LL, and MLH provided resources and helped perform the study. LL and JK helped analyze and interpret data. ACE drafted the manuscript and all other authors provided critical feedback for revision, and approved the final version.

## Competing interests
The authors declare no competing interests.
