## [Transparent Peer Review File · Communications Biology]

Reviewers' comments:

Reviewer #1 (Remarks to the Author):

This study by Ericsson et al is well conducted and relevant study showing that the microbiota that mice from different vendors harbor, drives behavioural phenotypes, when genetics are fixed. The strength of the paper is that the genetic background is fully fixed, because mice of the same stock harbors microbiota from four different breeders.

The paper is well written and should be published.

The paper could be modified in to a bit more humble attitude, because several other studies show that behavior is driven by the microbiota, and in this sense is not that surprising that when mice of uniform genetic background harbor four different microbiotas, you will get four different behavioral phenotypes. In line 30 of the abstract, it is said that 'Anecdotal evidence suggests that supplier-dependent GMs may influence behavior'. Instead of writing this I would write 'Several studies indicate that the microbiota drives behavior in mice, and therefore it is reasonable to hypothesize that mice of the same strain or stock from different vendors will have different behavioral phenotypes, as different colonies of mice are known to have different and colony-specific microbiotas'. I would elaborate a bit more on that in the introduction, and instead of spending an entire section in the discussion on infection studies, which seems a bit irrelevant here, I would relate my finding to other studies showing correlations between behavior and microbiota (and eventually also metabolism and behavior). The entire part in the discussion from line 305-312 about the MMRRRC consortium is very 'local' and not very relevant. Most readers would be able to understand that this study is relevant for the way we all cryopreserve and restore our strains, even if the reader never heard about MMRRRC.

In figure 1 D and E may be fused to one three-dimensional figure. It might be considered in all the figure to remove the horizontal lines under the significance letters, because they confuse the reader into thinking that only sex comparisons within a GM has been made, while it is in reality GM comparisons, which are shown.

Generally, the table do not look nice, and the layout should be improved prior to publication.

Reviewer #2 (Remarks to the Author):

The present study showed that embryonically derived CD1 pups fostered from surrogate dams of the 4 commercial mouse vendors displayed distinct differences in the richness of their fecal microiome associated with changes in animal behavior, physiological phenotypic parameters, glucose metabolism, and blood lymphocytes. The authors concluded that there is a profound influence of the different supplier-dependent gut microbiomes on host behavior and physiology in healthy, genetically similar wild type mice maintained in identical environments, and suggest that supplier-origin gut microbiomes represent potential sources of poor experimental reproducibility between labs. There are a few major concerns of the experimental design and the clarity of the writing in the manuscript:

Major points:

1. Abstract: the authors did not mention their key results on gut microbiome compositions or blood lymphocytes, which are essential components of the study.
2. Supplemental Figure 1: legend says n=12, figure says n=8. N=8 is too low for behavior tests. There should be at least 12-15 for mouse behavior tests.
3. The authors should avoid using the word GM (gut microbiome), since the 16S rDNA sequencing was only performed in the fecal microbiome.
4. Figure 1 and supplemental figures on microbiome: The authors should provide some specific

examples of distinct bacteria that are significantly altered among the 4 groups that serve as the driving force of the changes in alpha and beta diversity, in addition to showing these global changes.

5. Figure 1. The method section says n=12, the Figure 1 legend says n=5-6.

6. Method section, Mice paragraph, line 324-342: the authors should define other confounding factors, for example, which of the 4 CD1 colonies were housed in Columbia, which of the colonies were housed in UC Davis. Note the facility effect and the cage position effect (top vs. bottom) may be highly important confounding factors. Also the colonies housed in these two facilities are on different diet (LabDiet 5058 vs. Harlan Teklad diet 2918), which can tremendously impact gut microbiome independently from the authors' conclusions.

7. It is well-known that there are distinct host genetic differences among these 4 mouse strains, which may cause changes in embryonic development and/or microbiome independently from the authors' conclusions. Please critically evaluate the host genetics publication and discuss this.

8. The authors should examine the expression of host genes involved in glucose metabolism and immune response.

Minor points:

1. Abstract: line 33-34 is confusing: what was purchased from the suppliers, CD1 mice or surrogate dams? Define how many generations is "multiple generations"?
2. Figure 1 y axis: what is "ASVs"?
3. Method section: Please include the sample size for each experiment in addition to the sample size of the microbiome samples (n=12).
4. Line 418: spell out "SHIRPA".

Reviewer #3 (Remarks to the Author):

In this manuscript by Ericsson et. al., mice sourced from different vendors are compared for their gut microbiome (GM), behavior, morphology, and metabolism. A building body of work shows that differences in the housing and breeding practices can influence many downstream experimental parameters once the animals arrive at research institutions. Many of these changes appear to be driven by differences in the GM. Here the authors focus on how supplier GMs influence locomotor/movement, anxiety, body morphology, metabolism, cardiac function, and blood cell counts. The experimental strategy is to compare genetically identical CD-1 mice that have been rederived by embryo transfer into surrogate dams from the 4 major mouse suppliers in the U.S. and then maintained as separate colonies for >30 generations. After first convincingly showing that they are comparing 4 distinct gut microbiomes, the authors go on to conclude that mice with Envigo origin GM exhibit greater movement/exploration, reduced anxiety, smaller body size, and slower glucose uptake. No differences were noted in grip strength or vision.

This topic continues to be an important area that has broad interest in the research community. This study adds behavioral information, whereas most previous studies focused on how differences in GM changed the response to newly introduced pathogens. However, the manuscript is a general survey as opposed to going in depth into one or two parameters and showing that they impact physiology strongly enough that it would alter the conclusions made by an investigator within a particular experimental model system. The work would be significantly strengthened by the addition of one such example of this nature.

Specific comments:

1. Statistics are hard to follow within the figures. The legend often indicates lines above bars of data refer to significance, but it looks like only a subset of groupings are being compared. Values ideally should be listed within the legend. Sometimes two GMs seem to be grouped together (Figure 2A)- is the significance listed referring to all data points (P1-P4) collectively? When GM is indicated as significant determinant, this includes both male and female data points?
2. Why were CD-1 mice chosen for the study? Just historical due to good breeding or is there further rationale? Related, why were pseudopregnant C57BL6 mice chosen as surrogate dams in all cases but Charles River?
3. What if food intake was controlled for? Could this be done for an abbreviated time period to assess correlation with body size/metabolism? Without intake measurements any body size differences are hard to convincingly attribute to GM. This is acknowledged by the authors but works against the significance of the observations.
4. Were lymphocyte increases restricted to a particular subset? I don't see methods on how the data in Figure 6 was collected- flow cytometry?
5. Line 232-233: "These findings bring into question the interpretation of previous studies documenting behavioral and physiological differences between substrains from different suppliers." As noted above, a specific example demonstrating the significance of this problem due to GM would significantly strengthen the manuscript.

The authors sincerely appreciate the careful and thorough review of our manuscript. Individual reviewer comments are addressed below, with author responses in italic font. All line numbers referenced in our responses refer to the line numbers when viewing the revised manuscript with 'No Markup' in the Tracking tab.

Reviewer #1:

This study by Ericsson et al is well conducted and relevant study showing that the microbiota that mice from different vendors harbor, drives behavioural phenotypes, when genetics are fixed. The strength of the paper is that the genetic background is fully fixed, because mice of the same stock harbors microbiota from four different breeders. The paper is well written and should be published. The paper could be modified in to a bit more humble attitude, because several other studies show that behavior is driven by the microbiota, and in this sense is not that surprising that when mice of uniform genetic background harbor four different microbiotas, you will get four different behavioral phenotypes.

We have reviewed the manuscript and changed wording in places (e.g., line 75) to avoid overstating the import of our findings. If there are examples that were not addressed in our revised version, we're happy to make further edits.

In line 30 of the abstract, it is said that 'Anecdotal evidence suggests that supplier-dependent GMs may influence behavior'. Instead of writing this I would write 'Several studies indicate that the microbiota drives behavior in mice, and therefore it is reasonable to hypothesize that mice of the same strain or stock from different vendors will have different behavioral phenotypes, as different colonies of mice are known to have different and colony-specific microbiotas'.

The authors agree with the reviewer's suggested change in wording, and have amended that portion of the Abstract (lines 26-28) accordingly.

I would elaborate a bit more on that in the introduction, and instead of spending an entire section in the discussion on infection studies, which seems a bit irrelevant here, I would relate my finding to other studies showing correlations between behavior and microbiota (and eventually also metabolism and behavior).

*We appreciate the reviewer's suggestion and have expanded this portion of the Introduction slightly to include citations describing the behavioral and physiological differences observed in germfree mice (lines 54-61) and removed the citation (from the introduction) from Ivanov and Littman. However, we would respectfully contend that the wealth of data showing differential susceptibility to multiple infectious agents attributable to the fecal microbiota (e.g., *P. yoelli*, *Citrobacter*, *Enterobacteriaceae*, *C. Diff*) are very relevant in the context of the reported hematological differences. As such, we feel that these portions of the Discussion are very appropriate and would politely request to maintain them (albeit with requested edits).*

The entire part in the discussion from line 305-312 about the MMRRRC consortium is very 'local' and not very relevant. Most readers would be able to understand that this study is relevant for the way we all cryopreserve and restore our strains, even if the reader never heard about MMRRRC.

We agree with the reviewer's comment and have removed the text in question describing the MMRRC, in an effort to make the statements more broadly meaningful.

In figure 1 D and E may be fused to one three-dimensional figure.

*Per the reviewer's suggestion, the data shown in Figure 1, panels D and E are now presented in a single three-dimensional PCoA plot (revised **Figure 1D**).*

It might be considered in all the figure to remove the horizontal lines under the significance letters, because they confuse the reader into thinking that only sex comparisons within a GM has been made, while it is in reality GM comparisons, which are shown.

We appreciate the reviewer's comment and agree with their suggestion to remove the lines.

Generally, the table do not look nice, and the layout should be improved prior to publication.

We agree with the reviewer's assessment of the appearance of the Tables as presented. Tables were formatted per the journal guidelines. Should the manuscript be accepted, we will work with the editorial staff to make sure final versions are more aesthetically pleasing. To partially mitigate the problem, we have changed Tables to 1.5-line spacing.

Reviewer #2:

The present study showed that embryonically derived CD1 pups fostered from surrogate dams of the 4 commercial mouse vendors displayed distinct differences in the richness of their fecal microbiome associated with changes in animal behavior, physiological phenotypic parameters, glucose metabolism, and blood lymphocytes. The authors concluded that there is a profound influence of the different supplier-dependent gut microbiomes on host behavior and physiology in healthy, genetically similar wild type mice maintained in identical environments, and suggest that supplier-origin gut microbiomes represent potential sources of poor experimental reproducibility between labs. There are a few major concerns of the experimental design and the clarity of the writing in the manuscript:

Major points:

1. Abstract: the authors did not mention their key results on gut microbiome compositions or blood lymphocytes, which are essential components of the study.

We appreciate the astute observation and have added "hematological parameters" to the list of supplier-dependent differences detected in our study.

2. Supplemental Figure 1: legend says n=12, figure says n=8. N=8 is too low for behavior tests. There should be at least 12-15 for mouse behavior tests.

We understand and apologize for the confusion. Supplementary Figure 1 shows the timeline and sample sizes for the "experimental mice" which were $n = 8/\text{sex}/\text{GM}$, as depicted in the figure. We have added that parenthetically in the revised figure legend. The larger sample size ($n = 12/\text{sex}/\text{GM}$) refers to the microbiome data from parent colonies used as a comparator with the experimental mice to confirm that the mice had retained their distinct compositional characteristics throughout the study. Regarding the reviewer's concern with the sample size used, we would respectfully respond that sample sizes should be

based on the magnitude of the expected difference in relationship to the 'within group' variability. The large number of significant group-dependent differences identified in the current study indicate that our sample size was wholly sufficient. In the interest of the three R's of research, we do not feel there is adequate rationale for additional experiments or larger sample sizes.

3. The authors should avoid using the word GM (gut microbiome), since the 16S rDNA sequencing was only performed in the fecal microbiome.

We understand the reviewer's concern and used "gut microbiome" as jargon within the field, recognizing that it lacks specificity. To address this, the manuscript has been amended to use "fecal microbiome (FM)" throughout, and all figures have been amended similarly.

4. Figure 1 and supplemental figures on microbiome: The authors should provide some specific examples of distinct bacteria that are significantly altered among the 4 groups that serve as the driving force of the changes in alpha and beta diversity, in addition to showing these global changes.

*This is an excellent suggestion and the authors have revised **Figure 1** to include a new panel **E**, showing a Venn diagram listing the number of ASVs unique to each FM, or shared among multiple FMs. To provide a comprehensive summary of the differences in relative abundance between different FMs and sex, we have added a new **S3 Fig** and **S1 Table**. The new **S3 Fig** is a heatmap constructed from all ASVs yielding *p*-values below 0.05 associated with main effects of FM or sex, or interactions between the two factors, while **S1 Table** shows the *p* and *F* values associated with those tests. Together, these will allow readers to peruse the data and identify taxa associated with each group. Note that **S3 Fig** is a large file, but at an appropriate resolution that the relatively small text on the downloaded image can be magnified for viewing.*

5. Figure 1. The method section says *n*=12, the Figure 1 legend says *n*=5-6.

*With apologies for the confusion, these different sample sizes are in reference to either the number of fecal samples analyzed from the mice that were actually phenotyped (*n* = 5-6/sex/microbiome) or fecal samples from the parent colony (*n* = 12/sex/microbiome). We have tried to make this clear in the legends for panels **A** and **B** of **Figure 1**, and are open to additional suggestions.*

6. Method section, Mice paragraph, line 324-342: the authors should define other confounding factors, for example, which of the 4 CD1 colonies were housed in Columbia, which of the colonies were housed in UC Davis. Note the facility effect and the cage position effect (top vs. bottom) may be highly important confounding factors. Also the colonies housed in these two facilities are on different diet (LabDiet 5058 vs. Harlan Teklad diet 2918), which can tremendously impact gut microbiome independently from the authors' conclusions.

We have attempted to clarify in the Methods the details regarding where mice were housed and when samples were collected relative to shipping. As for confounding factors, we are unsure of the value of such comments, considering the high degree of agreement between the phenotyped mice and their parent colonies, despite all of those potential confounding factors. Additionally, we have published (and include the citation) studies demonstrating the conserved nature of these four microbiomes when shipped to various institutions, all with different husbandry procedures. If the reviewer could describe the context and placement of such comments, we would be happy to reconsider.

7. It is well-known that there are distinct host genetic differences among these 4 mouse strains, which may cause changes in embryonic development and/or microbiome independently from the authors conclusions. Please critically evaluate the host genetics publication and discuss this.

All of the mice tested in the current manuscript were CD-1 mice from the same supplier, and any host genetic differences between mice in the four colonies should be negligible. While these are an outbred stock, there is still limited allelic heterogeneity and we avoided filial matings within these colonies (and regularly introduced new host genetics via embryo transfer [ET]) to avoid fixation of alleles. While it is difficult to completely rule out the possibility of some subtle genetic factor influencing one of the tested microbiomes, the stark differences between those microbiomes would seem a much more plausible explanation. If the reviewer is suggesting that the genetic differences present in the original ET recipient/fecal microbiome donor mice (C57BL/6J, B6NTac, B6NHsd, and CD-1) are of relevance, we would remind the reviewer that these mice contributed no genetics to the mice tested in these studies as they served only as ET recipient surrogate dams, over 30 generations prior to the tested mice.

8. The authors should examine the expression of host genes involved in glucose metabolism and immune response.

Indeed, we are actively pursuing such investigations, but with limited dedicated funding. Thus, publication of these phenotypic differences will be used to support proposals to investigate both metatranscriptomic activity of these microbiomes, as well as host immune response at a finer resolution and transcriptional activity in several target organs (e.g., GI tract, developing CNS). As regards the current manuscript, such analyses are beyond the current budgetary constraints.

Minor points:

1. Abstract: line 33-34 is confusing: what was purchased from the suppliers, CD1 mice or surrogate dams? Define how many generations is “multiple generations”?

Based on multiple reviewer comments, we have reworded the Abstract, and removed that wording. We do describe later (in the Introduction) that these colonies were beyond the 30th generation when phenotyping was performed (although the exact generation is unclear).

2. Figure 1 y axis: what is “ASVs”?

We have revised these panels to read “Chao-1 index” on the y-axis. The Chao-1 index is the predicted richness (i.e., number of total amplicon sequence variants [ASVs]) in a sample.

3. Method section: Please include the sample size for each experiment in addition to the sample size of the microbiome samples (n=12).

We have clarified the sample sizes for FM analyses (lines 304 and 307) and phenotyping (line 353).

4. Line 418: spell out “SHIRPA”.

We have added the full name in front of the word SHIRPA in line 379.

Reviewer #3

In this manuscript by Ericsson et. al., mice sourced from different vendors are compared for their gut microbiome (GM), behavior, morphology, and metabolism. A building body of work shows that

differences in the housing and breeding practices can influence many downstream experimental parameters once the animals arrive at research institutions. Many of these changes appear to be driven by differences in the GM. Here the authors focus on how supplier GMs influence locomotor/movement, anxiety, body morphology, metabolism, cardiac function, and blood cell counts. The experimental strategy is to compare genetically identical CD-1 mice that have been rederived by embryo transfer into surrogate dams from the 4 major mouse suppliers in the U.S. and then maintained as separate colonies for >30 generations. After first convincingly showing that they are comparing 4 distinct gut microbiomes, the authors go on to conclude that mice with Envigo origin GM exhibit greater movement/exploration, reduced anxiety, smaller body size, and slower glucose uptake. No differences were noted in grip strength or vision.

This topic continues to be an important area that has broad interest in the research community. This study adds behavioral information, whereas most previous studies focused on how differences in GM changed the response to newly introduced pathogens. However, the manuscript is a general survey as opposed to going in depth into one or two parameters and showing that they impact physiology strongly enough that it would alter the conclusions made by an investigator within a particular experimental model system. The work would be significantly strengthened by the addition of one such example of this nature.

Specific comments:

1. Statistics are hard to follow within the figures. The legend often indicates lines above bars of data refer to significance, but it looks like only a subset of groupings are being compared. Values ideally should be listed within the legend. Sometimes two GMs seem to be grouped together (Figure 2A)- is the significance listed referring to all data points (P1-P4) collectively? When GM is indicated as significant determinant, this includes both male and female data points?

The authors apologize for the confusion. In response to these and other reviewer comments, the lines intended to connect the female and male samples has been removed from all figures. We have also endeavored to list all relevant p (and F) values in the Figure Legends, and clarify their meaning, as suggested. The reviewer's interpretation of the p values shown on figures is correct, in that these are always associated with FM-dependent differences, as opposed to sex- or time-dependent differences. To clarify this to readers, we have added a statement to relevant figure legends that the reported p and F values are associated with main effects of FM.

2. Why were CD-1 mice chosen for the study? Just historical due to good breeding or is there further rationale? Related, why were pseudopregnant C57BL6 mice chosen as surrogate dams in all cases but Charles River?

These colonies were initially generated by the MU MMRRC to investigate the influence of vendor-origin GMs on mouse physiology and model phenotypes distributed by the MMRRC. As such, these colonies were intended to be used as GM donors using embryo transfer and cross-fostering approaches, making a hearty outbred stock with favorable maternal behavior ideal. C57BL/6 substrains were used out of necessity, as Charles River is the only vendor providing CD-1 mice, and we believe that the background genetics of the initial pseudopregnant surrogate dams providing the fecal microbiota established in that colony are of minimal consequence at this point, as the microbiota of each colony has been 'passed' through dozens of generations of CD-1 mice.

3. What if food intake was controlled for? Could this be done for an abbreviated time period to assess correlation with body size/metabolism? Without intake measurements any body size differences are hard to convincingly attribute to GM. This is acknowledged by the authors but works against the significance of the observations.

This is an excellent idea, and we would like to perform follow-up studies to answer some of those questions. That being said, we believe that the reported differences are nonetheless attributable to the GM, and evidence of differences in food consumption would only help determine whether the differences in body size are due to increased intake or reduced activity, either one being mediated by differences in behavior. To perform such a study appropriately will require the use of metabolic chambers, and we do not currently have ready access to such platforms (but are investigating).

4. Were lymphocyte increases restricted to a particular subset? I don't see methods on how the data in Figure 6 was collected- flow cytometry?

The hematology was performed using a HemaVet Multispecies Hematology Analyzer HV950FS (Drew Scientific, CT, USA), a flow cytometer that provides FSC/SSC-based enumeration of the primary WBC subsets, but does not differentiate T cell subsets. In attempts to streamline the Methods, we provide references and links to all of the IMPC protocols, including the instrumentation used in the current study. Addition of one instrument would suggest listing of all instrumentation used in the phenotyping assays, which accumulates very quickly. We're happy to abide by reviewer and editorial direction, but personally prefer the current format. To provide the necessary information for an informed reader to understand the instrument used, we have added a brief description to line 510 (i.e., "using a clinical, flow cytometric hematology analyzer").

5. Line 232-233: "These findings bring into question the interpretation of previous studies documenting behavioral and physiological differences between substrains from different suppliers." As noted above, a specific example demonstrating the significance of this problem due to GM would significantly strengthen the manuscript.

The primary problem addressed by the data is the knowledge gap (and lack of awareness among the research community) regarding the influence of vendor-origin microbiomes on baseline behavior and physiology. We believe that the manuscript provides many examples in the literature reflecting or demonstrating this problem, including over a dozen manuscripts describing behavioral or physiological differences (directly related to what our data demonstrate) between substrains. In none of the referenced studies were mice co-housed or re-derived to control for the microbiota, and the microbiota is rarely even mentioned as a possible factor in all of those reported differences attributed to genetic divergence between the host substrains. The field of immunology and infectious disease is more appreciative of the microbiota as a factor and additional references (e.g., Denning et al. [ref 43] and Ivanov et al [ref 44]) describe studies where observed differences in substrain phenotype were pursued in the context of the microbiota, rather than host genetic features, to great effect. In summary, we feel that we have amply referenced the problem addressed by the current data, and believe that these data are of value to the research community in the context provided.

REVIEWERS' COMMENTS:

Reviewer #1 (Remarks to the Author):

In general, the authors have changed the manuscript according to my input, and I have no further to add. This is an interesting and important study.

Reviewer #2 (Remarks to the Author):

The authors have addressed all of my comments. No additional questions are raised.

Reviewer #3 (Remarks to the Author):

I appreciate the changes made in the revised version of the manuscript and I think these updates have improved clarity and representation of the data. I have only one comment:

1. New Figure S3 heat map- labels are unreadable. Could the X-axis in particular be improved so that we can see how the groups are clustering? Or an additional larger bracketed 'group label' added? Understood that this is a large file and problematic to work with, but I can't tell what is being shown.

College of Veterinary Medicine

University of Missouri-Columbia
An Equal Opportunity /ADA Institution

Department of Veterinary Pathobiology

Aaron Ericsson, DVM, PhD

4011 Discovery Dr.

Columbia, MO 65201

Office: (573) 882-1019

Fax: (573) 882-9857

Email: Ericssona@missouri.edu

May 15, 2021

To whom it may concern,

We sincerely appreciate the thorough and constructive reviews provided by the reviewers. We have amended the heatmap in Supplementary Figure 3 to improve its readability. Please do not hesitate to contact us if additional changes are required.

Best regards,

Aaron Ericsson, DVM, PhD

Assistant Professor

Director, University of Missouri Metagenomics Center